# Unmanned Ariel Vehicle (UAV) Path Planning for Area Segmentation in Intelligent Landmine Detection Systems

**DOI:** 10.3390/s23167264

**Published:** 2023-08-18

**Authors:** Ahmed Barnawi, Krishan Kumar, Neeraj Kumar, Nisha Thakur, Bander Alzahrani, Amal Almansour

**Affiliations:** 1Faculty of Computing and Information Technology, King Abdulaziz University, Jeddah 21589, Saudi Arabia; neeraj.kumar@thapar.edu (N.K.); aalmansour@kau.edu.sa (A.A.); 2Thapar Institute of Engineering and Technology, Deemed to be University, Patiala 147004, India; kkumar_phd19@thapar.edu (K.K.); nthakur_phd20@thapar.edu (N.T.)

**Keywords:** landmine detection, segmentation, coverage path planning, deep learning

## Abstract

Landmine contamination is a significant problem that has devastating consequences worldwide. Unmanned aerial vehicles (UAVs) can play an important role in solving this problem. The technology has the potential to expedite, simplify, and improve the safety and efficacy of the landmine detection process prior to physical intervention. Although the process of detecting landmines in contaminated environments is systematic, it is proven to be rather costly and overwhelming, especially if prior information about the location of the lethal objects is unknown. Therefore, automation of the process to orchestrate the search for landmines has become necessary to utilize the full potential of system components, particularly the UAV, which is the enabling technology used to airborne the sensors required in the discovery stage. UAVs have a limited amount of power at their disposal. Due to the complexity of target locations, the coverage route for UAV-based surveys must be meticulously designed to optimize resource usage and accomplish complete coverage. This study presents a framework for autonomous UAV-based landmine detection to determine the coverage route for scanning the target area. It is performed by extracting the area of interest using segmentation based on deep learning and then constructing the coverage route plan for the aerial survey. Multiple coverage path patterns are used to identify the ideal UAV route. The effectiveness of the suggested framework is evaluated using several target areas of differing sizes and complexities.

## 1. Introduction

Landmines are devices with explosive properties that are strategically placed on the ground or buried slightly beneath the surface. Their purpose is to harm or incapacitate adversary assets, including troops, vehicles, or tanks, as they approach or pass over the mine. The mines can be categorized into two types: anti-personnel and anti-vehicle, based on their intended targets. The detonation is triggered by the pressure, presence, or proximity of a person or vehicle [1]. Landmine explosions can cause severe consequences and physical damage.

Landmines are indiscriminate weapons with unfortunate far-reaching consequences even after conflicts cease. They bear significant social, economic, and environmental implications, serving as area-denial munitions and tactical barriers [2]. While their deployment is relatively cheap and easy, removing them is time-consuming, hazardous, and costly. Figure 1 displays the casualties from landmines or Explosive Remnants of War (ERW) recorded by Landmine Monitor for 2015–2021. According to a 2022 report [3], sixty countries and territories remain contaminated by anti-personnel landmines, with fifty casualties reported in 2021. Of the 5544 fatalities in 2021, 4200 were civilians.

Landmine clearance or demining refers to removing landmines from a particular location. There are both military and humanitarian demining operations. Military demining aims to quickly clear a passage through a minefield, whereas humanitarian demining removes all landmines to a set depth and makes the area safe for human usage. Considering the catastrophic effects of landmines, measures must be undertaken to clear the minefields. Landmine clearance is a challenging and time-consuming task. It is a high-risk operation requiring specific equipment and specialized human resources to locate and remove landmines. In 2021, there were a total of 27 reported casualties of deminers [3].

As unmanned aerial vehicles (UAVs) and sensor technologies advance, they are finding more valuable use cases. The aerial survey using drone technology has the potential to save many lives by making the clearance of landmines a safer, quicker, and more affordable process. UAVs also make it feasible to survey otherwise inaccessible and complex geographies. The minefield can be inspected using a UAV mounted with suitable sensor devices for detecting landmines. The UAVs’ resources, such as their batteries, are limited. Additionally, the operation involves a complicated environment with obstructions. It necessitates determining the best UAV traversal route that covers the whole target area.

The detection of landmines poses a significant challenge since numerous elements influence the accuracy and efficacy of this operation.

Sensor technology: The selection of sensor technology plays a crucial role in the detection of landmines. Various types of sensors, including metal detectors, ground-penetrating radars (GPRs), and magnetometers, exhibit diverse capabilities and sensitivities toward certain categories of landmines [4]. Every type of sensor possesses distinct advantages and limits when detecting particular landmine materials and depths of burial.Landmine type: Landmine detection is influenced by their size and material composition. Detecting larger or metallic landmines is typically more feasible than smaller or non-metallic landmines. The capacity of a sensor to identify landmines with varied compositions is contingent upon its sensitivity to various materials.Soil type: The detection of landmines is influenced by several soil conditions, including but not limited to moisture content, conductivity, and magnetic susceptibility, as these soil properties play a significant role in the detection process. Variations can influence the electrical conductivity of the soil in moisture levels, impacting the sensor’s performance. The presence of diverse soil compositions and changes in soil types can potentially lead to erroneous alerts or instances of undetected occurrences.Environmental factors: Environmental factors, including nearby structures or electronic devices emitting electromagnetic interference, can introduce unwanted signals into the sensor data, impacting its accuracy and dependability. Precise calibration and signal processing techniques must be meticulously applied to counteract this interference.Spatial resolution: The spatial resolution of the sensor system, which is determined by the sensor’s characteristics and flight parameters, influences its ability to detect small or closely spaced landmines. A higher spatial resolution can increase detection accuracy but may reduce the flight’s coverage area.Flight altitude and speed: The UAV’s altitude and speed influence the efficacy of the sensor. Higher altitudes may reduce spatial resolution, while lower altitudes may provide greater sensitivity but increase the risk to the UAV. The flight pace influences the time available for data collection and may affect the data quality.Terrain and topography: The UAV’s stability and the sensor’s proximity to the ground might be influenced by the topography and roughness of the terrain. Variations in readings may occur due to the influence of uneven terrain on the orientation of the sensor.

Utilizing UAV technology to conduct low-altitude, low-speed surveys for landmine detection is of the utmost importance. It takes advantage of the UAV’s ability to operate close to the ground, thereby facilitating the acquisition of detailed data and enhancing the detection of smaller or more deeply concealed landmines. The method with reduced survey speeds ensures comprehensive data acquisition, improving target identification and discrimination capabilities.

This research focuses on the challenge of automated area segmentation and route planning for UAV-based landmine detection. Initially, deep learning-based extraction isolates the region of interest (ROI) from the satellite imagery. Then, a route determination strategy for UAV-based coverage is formulated. It helps find the most efficient route considering both time and power consumption. UAV uses the specified course to traverse the target region. When equipped with suitable sensing technologies, the signatures of landmines can be recognized and localized. It paves the way for automated, remote landmine-detecting operations. In our study, we leverage offline path planning as a fundamental component of the proposed approach for UAV-based aerial surveys for landmine detection. Its primary objective is to generate an optimized coverage path that the UAV will follow during its survey mission. Once the path planning is completed, the optimized coverage path is transferred to the UAV’s onboard system. The generated path guides the UAV’s autonomous execution during the aerial survey. It enables the UAV to collect sensor data systematically and efficiently, improving data quality. Moreover, the streamlined path minimizes energy consumption, allowing extended flight endurance and maximizing the survey area coverage. The onboard computations primarily pertain to real-time navigation and control, ensuring that the UAV accurately follows the predetermined path.

### 1.1. Motivation and Contribution

As a result of this paper, the following notable contributions are made.

Deep learning-based segmentation is used to extract the region of interest (ROI) from satellite imagery of the geographical area.The path planning strategy is formulated to optimize the coverage route of the UAV-based region survey.The workflow is designed for UAV-based minefield surveys.The efficacy of the proposed strategy is evaluated in terms of energy and time constraints.

### 1.2. Organization

The general outline of the paper is as follows. Section 2 provides an overview of segmentation and path planning concepts. In Section 3, we dive deeper into the details of the proposed framework. In Section 4, the results and their interpretation are presented. Section 5 provides a conclusion of the work and suggests some avenues for further study.

## 2. Background

Unmanned aerial vehicles (UAVs), sometimes known as “drones”, are unmanned aircraft that can be flown without a pilot on board. Aircraft, ground control stations, and communications systems all fall under the umbrella term unmanned aircraft systems (UAS), which describes the infrastructure necessary for sophisticated drone operations. An autonomous drone is a UAV that can fly missions independently of a human pilot. It can take off, execute its task, and return to base without human assistance. Rather than relying on a human pilot, communications management software handles mission planning and flight control for autonomous drones.

### 2.1. UAV-Based Landmine Detection

Major application areas for drone technology include agriculture, disaster management, remote surveying, network coverage, and product delivery. In addition to the developments in network technology, UAVs play a greater role in maintaining and enhancing network capabilities. The UAVs play a crucial role in topographical surveying by offering a quick, simple, inexpensive, and safe approach with decreased human participation and improved access to challenging terrain. Recent research has envisioned a similar system for landmine detection [4]. Various sensing devices can detect the landmine signature, including metal detectors, ground penetration radar (GPR), infrared, multispectral, acoustic, magnetometer, and others [2,5]. More specialized sensors are now compatible with UAVs, allowing for more precise and thorough airborne surveys as technology advances.

Figure 2 depicts the process of landmine detection where the minefield survey utilizes a UAV equipped with detection sensors. The area is first chosen for the UAV-based investigation and its geographical map serves as the input. The required portion is then retrieved to determine the region of interest. A route or path plan is established to produce the coverage path in the extracted area. UAVs utilize the created course to explore all geographical places and gather sensor-based information. Information collected may subsequently be processed using artificial intelligence (AI) techniques to help locate and identify landmines. Additionally, the minefield layout may be recognized by post-processing. The demining team can use the detection results to inspect the specified locations and remove or destroy the landmines.

The use of UAVs in landmine detection provides several benefits, including a reduction in survey time, improved accessibility to difficult terrain, and risk-free operation. Consequently, several researchers have investigated the possibility of improving the detection procedure by using UAVs. Using low-cost drones, the visual detection system was proposed in [6] to detect fully and partially visible landmines. The system provides good efficiency in a low-altitude survey and low flight speed. The UAV equipped with a metal detector was considered in [7] to carry out aerial detection in rough terrains. The developed system can identify metals, mines, and bare explosives buried underground. A metal detector requires a certain amount of metal content to generate the landmine signature. It makes the detection of low-metal landmines difficult using metal detectors. Colorado et al. [8] integrated a software-defined radio (SDR)-based GPR system with a UAV that was capable of detecting landmines in variable terrains. The flight control system was developed to enable the steady operation of the UAV. The ground-penetrating synthetic aperture radar (GPSAR) was used with UAV in [9] to accelerate the demining process. The strong clutter generated at the surface–air interface affected the GPR-based landmine detection. The time-gating and average subtraction technique was used for clutter removal in [10], and SAR processing was used to obtain high-resolution images.

A lightweight and low-power GPR was developed in [11], based on the stepped frequency continuous wave (SCFW) radar. The developed system efficiently detects metallic and plastic landmines while flying at low altitudes. But the detection system has limited scanning speed and suffers from the moisture content of the soil. In [12], an improved SAR-based GPR detection system was used with UAV for landmine and improvised explosive device (IED) detection. The system provides a better signal-to-clutter ratio in GPR images using clutter filtering based on singular value decomposition (SVD).

The UAV equipped with a thermal and multispectral sensor was employed for scatterable landmine detection in [13]. It is based on a convolution neural network (CNN)-based identification and localization of landmine signatures in the captured sensor images. The magnetometer-based survey using UAV was presented in [14] to detect different types of mines. The study has taken into account a sensor-equipped UAV with a height of 1 m and a survey line spacing of 3 m. It was improved in [15] by altering the sensor position to enable low-altitude operation. The low-pass filtering and moving average methods were used to eliminate the magnetic noise and improve detection.

Various sensing technologies can be used with UAVs for the remote survey of minefields, each serving some advantages and disadvantages in different use cases. In most studies, the survey was remotely controlled in a simple environment. It poses a challenge for UAV operation in complex terrains where the optimal utilization of resources and complete coverage are critical. It requires automation of trajectory formulation for UAV-based surveys.

### 2.2. Segmentation

Segmentation is the process of breaking down a picture into multiple pieces or regions [16]. It entails dividing a picture into smaller, more manageable regions, with each section being represented by a mask or a set of labels. Simply said, it is the process of assigning labels to the image pixels. The pixels that share the same attributes are labeled alike. It helps in the identification of different image components, and instead of processing the complete image, just the relevant parts need to be processed. The segmentation can be categorized as semantic segmentation, instance segmentation, and panoptic segmentation.

In this study, K-means clustering and normalized RGB color space are used to segment the green terrain in the provided image. Using K-means clustering and normalized RGB color space, the results demonstrate that the shared Google Earth photographs can be divided into segments that are roughly 40.50% and 47.01% of the total image pixels, respectively [17].

When image segmentation methods are developed, they could enter the medical area and help with disease diagnosis. Using pathology images to expedite clinical diagnosis and automate image analysis with reliability and effectiveness is still highly challenging. For several machine learning methods, including convolutional neural networks, automatic pathological picture segmentation was proposed. Accuracy and processing speed are provided for this segmentation using fully convolutional networks and other deep learning techniques [18].

The segmentation process plays a crucial role in enhancing the efficacy of landmine detection procedures. By precisely identifying and isolating areas with a high probability of containing landmines, segmentation lays the groundwork for targeted and efficient UAV-based aerial surveys. Moreover, the segmentation process assists in overcoming the obstacles posed by complex and diverse terrains. It enables the UAV to prioritize survey activities, concentrating on pre-identified regions more likely to contain landmines. This targeted strategy not only increases survey efficiency but also improves operational safety. By incorporating the segmentation process into the landmine detection workflow, the accuracy, efficiency, and overall effectiveness of UAV-based aerial surveys can be significantly improved. By intelligently identifying regions of interest, the UAV can implement survey missions with greater precision, resulting in enhanced landmine detection results and the establishment of safer environments. As part of our investigation, we collected satellite images from Google Earth for our intended use. The objective is to determine the precise target area for UAV aerial surveys. This requires the initial collection of datasets, followed by the precise definition of the intended area of interest. Then, a deep learning model is utilized to effectively recognize and categorize the distinct regions.

### 2.3. UAV Path Planning

In recent years, various studies are focused on automating the UAV flight. One such component examined in the literature to automate the UAV-based aerial survey is autonomous route determination, also known as path planning. It covers the estimation of the best route between a source and a destination [19]. In some applications, like archaeological surveys, the objective is to cover every location within a region. It requires the determination of the optimal coverage path, denoted as coverage path planning [20]. Figure 3 illustrates the execution of an aerial survey, wherein the UAV systematically traverses a predetermined coverage path to collect data, covering the whole target region.

Several recent works have investigated the coverage path planning of UAVs. Surveys involving UAVs encompass operations under a range of conditions, accounting for varying complexities and the number of regions Accordingly, different strategies have been formulated to automate the coverage path determination. The target region having complex geometry often requires partitioning the free space into cells to simplify the survey path, termed cellular decomposition [21]. It can be categorized as the exact and approximate cellular decomposition or grid decomposition. In the exact decomposition method, the region is partitioned into smaller sub-areas, the combination of which forms the same region. The path for each sub-region is then determined, and the integration of all the paths generates the coverage path. On the other hand, the approximate decomposition discretizes the area into a set of regular cells at some resolution. The traversal of all the cells in some order generates the coverage path.

The authors of [22] focused on the exact cellular decomposition of the region for a UAV-based survey using a greedy strategy. The back-and-forth motion was used to obtain the coverage path for each sub-region. The minimum spanning tree (MST)-based traversal was used to obtain the complete coverage path for the undirected graph of sub-regions. Torres et al. considered the line sweep direction computation for the polygon, and then the coverage path was generated perpendicular to it. In [23], the work focused on UAV operations in the presence of wind, and a dynamic programming-based approach was used for polygon decomposition. The selected direction of the coverage path line was perpendicular to the wind direction to consume the minimum flight time.

The approximate cellular decomposition and a gradient-based approach were used in [24] to obtain a coverage path with the minimum number of turns. The energy model for UAV was derived in [25], which was used to determine the optimal coverage path along with safety mechanisms. The authors of [26] examined the coverage path formulation based on exact and grid-based decomposition methods for various CPP widths.

In some applications, the target area to be surveyed can be considerably large, making it difficult to be covered with a single UAV. Also, certain applications, like disaster management, are time-critical and require the survey to be completed as quickly as possible [27]. The survey task can be enhanced by employing multiple UAVs leading to cooperative or multi-UAV coverage [28]. The region can be decomposed into smaller fragments that are assigned to different UAVs. It requires careful partitioning and sub-region allocation to optimize the complete coverage [29]. The hexagonal decomposition of the target region and clustering-based allocation to multiple UAVs was proposed in [30]. Similarly, Morse-based decomposition was used in [31] where sub-region assignment considered the UAV start and end positions. Choi et al. [32] proposed an optimization model with column generation to track energy consumption during an aerial survey. The work focused on minimizing the number of UAVs and energy requirements for aerial surveys.

Researchers have investigated the cooperative utilization of several UAVs to achieve sustained coverage, with a focus on grids that are visited less frequently. Furthermore, specific strategies have been implemented to address emergency situations in particular regions. In some scenarios, multiple UAVs can operate in a certain formation, and the coverage path can be determined accordingly [33]. The authors of [34] included mixed-integer linear programming (MILP) modeling of the coverage task and proposed a randomized search-based algorithm to optimize the coverage path for heterogeneous UAVs.

In certain applications, UAVs need to survey multiple separated regions instead of a single area. It requires proper planning and sequence determination if a single UAV is used to cover all the regions to determine an efficient coverage path. Xie et al. [35] formulated the problem as the integration of the traveling salesperson problem (TSP) and CPP. Grid-based and dynamic programming (DP)-based approaches have been used to determine each region’s entry and exit point and optimize intra and inter-regional coverage paths. The work was extended in [36], where a heuristic method was employed. The exact methods have limited scalability, where the time complexity increases exponentially with the problem scale. The heuristic approaches can provide appropriate solutions in an acceptable time and improved scalability.

In UAV-based landmine detection, the minefield survey often includes complex terrain, and the UAV resources are also constrained. The survey also needs to consider several factors to determine the gap between sweep lines. It, thus, requires autonomous and efficient path planning that determines the route for UAV passing through each location of the target region with minimum energy and time requirements. The proposed work is focused on addressing the CPP problem for landmine detection.

## 3. Methodology

The determination of the coverage path for a UAV-based survey includes two phases: ROI segmentation and path planning. Figure 4 depicts the various steps involved in the whole process. Initially, a geographical map serves as input for selecting a target area. The region is segmented using a deep learning-based technique to extract the area of interest. The UAV route is created as a series of geographical waypoints in the subsequent path planning stage using the coverage path determination and coordinate conversion strategies.

### 3.1. ROI Segmentation

The first phase involves extracting the region to be surveyed from the input region map. Landmine detection is often carried out in terrains where every part or region, like rocky regions, does not form a part of detection. So, it becomes essential to identify the region of interest so that only desired area is considered for the UAV-based survey. The input to the first phase is in the form of a satellite map image of the location where landmine detection is to be carried out. The ROI is segmented from the image, acting as an input for the path planning phase.

The U-Net model is used for image segmentation that segments the input satellite image into ROI and non-ROI regions. The ROI region is the intended ROI that the UAV surveys for detecting landmines. U-Net is a classic image segmentation model used in a wide range of applications. It has two networks: an encoder and a decoder. U-Net conducts semantic segmentation that is based on pixel-level classification. Due to several factors, U-Net has increased in popularity and is seen as superior to several other semantic segmentation techniques.

Satellite images: U-Net can be used to segment various land cover types, including forests, aquatic bodies, urban areas, and agricultural land, in satellite imagery. The model will produce masks with separate labels for each land cover type. And here, our data takes satellite images of a particular region.The U-Net architecture is made to handle small amounts of data: Obtaining big annotated datasets for training deep learning models can be difficult in many real-world applications. Due to its symmetric design and skip connections, U-Net can function well even with little training data. Here, we captured satellite images that are limited in number, so we used the U-Net model for image segmentation.

Figure 5 shows the encoder in its first half. It involves the usage of a pre-trained classification network like VGG/ResNet. Convolution blocks and max pool downsampling encode the input image into feature representations at different levels. The second element of the architecture is the decoder. It semantically projects the encoder’s learned and low-resolution selected features onto the higher-resolution pixel space to provide a dense categorization. Upsampling, concatenation, and conventional convolution procedures make up the decoder. It consists of a route that expands to the right and contracts to the left. Here we took the input size of an image (256, 255, 3). The contracting path follows the common convolutional network architecture. Repeatedly, a rectified linear unit (ReLU) and a 2 × 2 max pooling operation with stride 2 are used for downsampling, followed by two 3 × 3 convolutions (unpadded convolutions). With each downsampling step, we double the number of feature channels by four. A 2 × 2 convolution (“up-convolution”), which reduces the number of feature channels in half, an association with the correspondingly cropped feature map from the contracting path, and two 3 × 3 convolutions, each followed by a ReLU, at each stage of the expansive path are all performed after upsampling the feature map. Cropping is necessary because each convolution loses boundary pixels. Using a 1 × 1 convolution, each 64-component feature vector is translated to the necessary number of classes at the top layer. There are a total of 23 convolutional layers in the network. An associated multi-channel feature map is shown by each blue box. On the top of the box, there is a channel count indication. In this case, brown squares signify replicated feature maps. The various operations are shown by the arrows.

### 3.2. Path Planning

#### 3.2.1. Coverage Path for ROI

The second phase includes the formation of a UAV trajectory that covers all the points of the segmented region obtained from the previous stage. This paper examines the use of spiral and zig-zag algorithms to determine an optimal coverage path for the UAV employed in aerial surveys for landmine detection. The spiral path planning algorithm was selected for its ability to efficiently cover extensive areas while maintaining a central focus, which corresponds well with the requirement for comprehensive coverage in landmine detection. In contrast, the zig-zag path planning algorithm provides a structured approach for systematic coverage. It is selected due to its unique ability to minimize the number of turns and overall path length, which are essential for optimizing the efficacy of UAV-based aerial surveys. The coverage path is affected by region shape, sensor footprint, UAV height, resolution requirement, and other factors. All the parameters determine the waypoints and distance between survey lines. For the sake of simplicity, our proposed approach entails generating a generalized path that adheres to the predetermined line gap and UAV height specifications. This streamlined approach facilitates ease of implementation and serves as a foundation for more complex adaptations in the future.

Energy efficiency is vital for UAV missions, as it directly impacts the UAV’s endurance and operational range. Minimizing the path length helps conserve energy and extend the mission’s flight time, enabling the UAV to cover larger areas or stay airborne for extended periods. Turning points in the flight path of a UAV during aerial surveys also play an essential role in determining the mission’s efficacy and energy consumption. A turning point is where the UAV alters direction or bearing to navigate the survey area. Depending on the path planning strategy, these locations may be abrupt turns or incremental course adjustments. Turning points require the UAV to adjust its orientation and alter its flight trajectory. Each turn consumes additional energy because the UAV must surmount aerodynamic drag and inertial forces during the direction change. Frequent reversals can increase energy consumption, reducing the mission’s overall endurance and flight range. The time the UAV takes to complete its mission increases with every turn. The length of a mission is affected by factors like the time needed to complete a turn and the time needed to stabilize the UAV’s flight after the turn. Keeping the number of turns to a minimum improves trip time and data-gathering efficiency.

The primary objective of path planning is to conserve energy and time by shortening the path and minimizing turns. It is accomplished by determining the most feasible path that avoids needless detours and abrupt turns. With this optimized route, the UAV can efficiently survey the area, consuming less energy and concluding the mission faster. It makes UAV missions more cost-effective and sustainable while maximizing resource utilization and enhancing overall performance.

The spiral and zig-zag path planning methods reduce the turning radius. They frequently provide tighter turns and lower turning radii, which might be essential in confined spaces. It may result in more significant space usage and more effective navigation. Our dataset contains satellite images of particular regions, with ROI of different polygon shapes with tighter turns and less turning radii. These path planning methods can help minimize the risk of collision. These algorithms also provide smooth paths that can increase energy efficiency since unexpected shifts in direction can increase energy consumption, particularly for mobile robots or vehicles.

Algorithm 1 describes the procedure for spiral path planning. First, we initialize the inputs with $S_{I}$ representing the starting point and $R_{I}$ representing the region of interest. It then starts with the spiral path planning method, which divides the ROI into grid form. Here, we have considered the width of size *w* for each grid.
**Algorithm 1** Spiral path planning (Spp) algorithm**Input**: $S_{I}=Starting\;point,$$R_{I}=Region\;of\;interest$w=LinewidthOutput:Waypoints:N **Procedure**:
1:  $N\leftarrow S_{I}$2:  $r_{s}=0,r_{e}=m-1$3:  $c_{s}=0,c_{e}=n-1$4:  **while**
$\left(\left(r_{s}<=r_{e}\right)\;\&\&\;\left(c_{s}<=c_{e}\right)\right)$
**do**5:       **for** $\left(i=c_{s};i<=c_{e};i=i++\right)$ **do**6:             $N\leftarrow (i,r_{s})$7:             $r_{s}=r_{s}+1$8:             **for** $\left(j=r_{s};\;j<=r_{c};j=j++\right)$ **do**9:                   N←(j,i)10:                 **if** $\left(r_{s}⩽=r_{e}\right)$ **then**11:                     **for** $\left(k=c_{e};\;k>=c_{s};k=k--\right)$ **do**12:                           N←(k,j)13:                           $r_{e}=r_{e}-1$14:                           **if** $\left(c_{s}<=c_{e}\right)$ **then**15:                               **for** $\left(l=r_{e};l>=r_{s};l=l--\right)$ **do**16:                                     N←(l,k)17:                                     $c_{s}=c_{s}+1$18:                                **end for**19:                           **end if**20:                      **end for**21:                 **end if**22:           **end for**23:      **end for**24:**end while**       **return**  N

The explanation of the algorithm is given below:Lines 2, 3 initialize variables $r_{s}$, $r_{e}$, $c_{s}$, and $c_{e}$, which represent the start and end indices of the rows and columns, respectively, stored in the current iteration.In line 4, it enters an iteration that continues as long as the start index of the rows is less than or equal to the end index of the rows and the start index of the columns is less than or equal to the end index of the columns.Lines 6, 7 use iteration to store the elements in the top row from left to right. The start index of the rows is incremented by 1 to indicate that the top row has been stored.In line 8, it again goes into another iteration to store the elements in the rightmost column from top to bottom. The end index of the columns is decremented by 1 to indicate that the rightmost column has been stored.In line 10, it checks if there are any rows left. If so, in line 11, it again goes into iteration for the elements in the bottom row from right to left. The end index of the rows is decremented by 1 to indicate that the bottom row has been stored.Line 14 checks if there are any columns left. If so, it uses an iteration again for the elements in the leftmost column from bottom to top.The start index of the columns is incremented by 1. The iteration continues with the updated indices, and the process is repeated until all elements are stored in ordered nodes (N).It then returns the coordinates in an ordered node N.

Figure 6 shows the spiral path planning on the segmented image. The complexity of the algorithm is O(r*c). It is explained below:The algorithm uses nested loops. The outer while loop continues until both $r_{s}$ are less than or equal to $r_{e}$ and $c_{s}$ are less than or equal to $c_{e}$.Inside the while loop, there are two nested for loops:–The first nested for loop iterates over rows within the current column (from $r_{s}$ to $r_{e}$).–The second nested for loop iterates over columns within the current row (from $c_{s}$ to $c_{e}$).Additionally, there are two more nested for loops within conditions:–If $r_{s}$ <= $r_{e}$, a loop iterates over a row in reverse (from $c_{e}$ to $c_{s}$).–If $c_{s}$ <= $c_{e}$, a loop iterates over a column in reverse (from $r_{e}$ to $r_{s}$).In terms of time complexity:–The first while loop runs at most min(r,c) times, where r is the number of rows and c is the number of columns.–The nested loops inside the while loop are responsible for iterating through submatrices with decreasing dimensions. The total number of iterations performed by these loops can be roughly approximated as O(r*c).

Therefore, the overall time complexity of the algorithm is approximately O(r*c).

**Figure 6 sensors-23-07264-f006:**
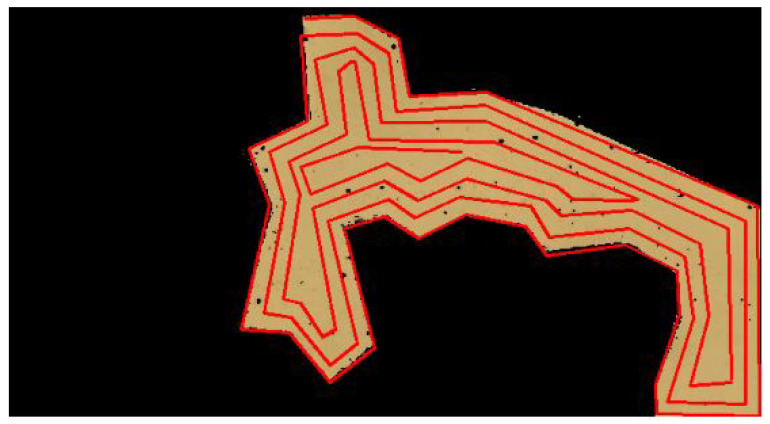
Spiral path planning.

Algorithm 2 depicts zig-zag path planning. It takes starting point $S_{I}$, R is the ROI of an input image, $r_{e}$ is the even row, $r_{o}$ is the odd row, c is the column, and in N, we store the waypoints. It outputs a list of ordered nodes (N), which represents a path of points in the ROI R.
**Algorithm 2** Zig-zag path planning**Input**: $S_{I}$: Starting point *R*:ROI Linewidth:w **Output**: Waypoints N **Procedure**:
1:  $N\leftarrow S_{I}$2:  $r_{e}=0$3:  $r_{o}=1$4:  **while** $r_{e}<r$ **do**5:        **for** iinrangec **do**6:             $N\leftarrow (i,r_{e})$7:             $r_{e}=r_{e}+2$8:             **if** $r_{o}<r$ **then**9:                **for** jinrange(c−1,−1,−1) **do**10:                   N←(j,i)11:                   $r_{o}=r_{o}+2$12:              **end for**13:          **end if**14:     **end for**15:**end while**       **return** N

It then starts with the zig-zag path planning method, which divides the ROI into grid form. We took the linewidth of size w for each grid. Here is a step-by-step explanation of the procedure:Line 1 points to the starting point and stores it in N. It then initializes variables $r_{e}$, and $r_{o}$ to 0 and 1 respectively (lines 2,3).Then it starts with the first iteration (lines 4,5), where it starts with the even row and stores the waypoints of the path in the same direction in N.We skip the next row (oddRow) and move to the next evenRow.Similarly, it again iterates (line 8) for the odd rows and stores the path waypoints in the opposite direction in N.We skip the next row and move to the next even row.Finally, we obtain the output in the form of a “path” list as the ordered nodes(N) of the path for the ROI R. Note that this procedure traverses the ROI, R, in a zig-zag pattern, alternating the direction of traversal in each row.

Figure 7 shows the zig-zag path planning on the segmented image. The computation complexity of the above algorithm is explained below:Initializing N with the values of $S_{1}$ takes constant time, O(1).Initializing variables $r_{e}$ and $r_{o}$ takes constant time, O(1).The while loop continues as long as $r_{e}$ is less than r. Each iteration involves a constant number of operations (incrementing $r_{e}$ and potentially entering the inner loop). The number of iterations is r/2, so the complexity of the while loop is O(r).The first for loop iterates over the indices of array $r_{e}$, which contains c elements. Each iteration involves a constant number of operations (assigning a value to N and incrementing $r_{e}$). So, the complexity of this loop is O(c).The second for loop (inside the first loop) iterates in reverse over the indices of array $r_{o}$, which contains c elements. Similar to the previous loop, each iteration involves a constant number of operations. Since this loop only occurs if $r_{o}$ < r, we can ignore its complexity if $r_{o}$ is small compared to r.

**Figure 7 sensors-23-07264-f007:**
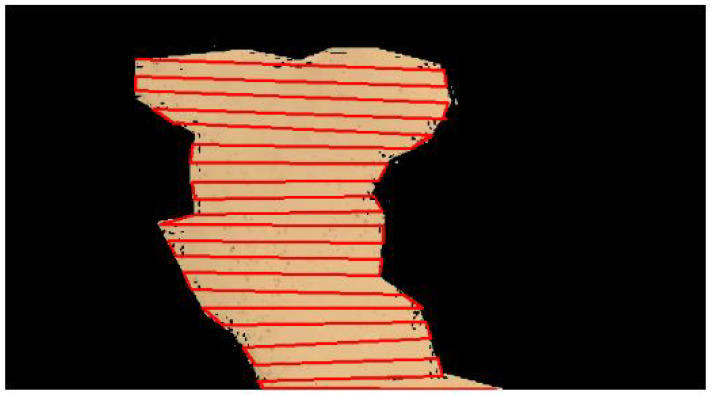
Zig-zag path planning.

The overall time complexity is determined by the most time-consuming part of the code, which is the while loop. Therefore, the total time complexity of the provided pseudo-code is approximately O(r * c), where r is the predefined value and c is the number of elements in arrays $r_{e}$ and $r_{o}$.

#### 3.2.2. Coordinates Transformation

The result of the path planning is a sequence of waypoints representing the order of locations in the path to be followed by UAVs. Since the segmentation’s input and output are in the form of an image, the waypoints are also obtained in pixel coordinates. The transformation of pixel coordinates to geographical coordinates is required for UAV operation. To achieve it, the geographical coordinates of some points, like corner pixels, are maintained and later used for mapping pixel coordinates with geographical ones. In this way, the final coverage path as a sequence of geographical coordinates is obtained.

## 4. Experimentation and Results

The proposed method, including segmentation and path planning, was simulated using Python and the ArduPilot-based Mission planner simulation. The goal is to obtain the segmentation accuracy and efficiency of generated UAV coverage path.

### 4.1. Segmentation

#### 4.1.1. Dataset

The dataset is obtained from Google Earth, where 100 samples are taken from various places. The Label box tool is employed to assign labels to the pixels of each instance, categorizing them into two distinct classes: region of interest (ROI) and non-ROI. Figure 8a,b contains the original image, and Figure 8c,d contains the segmented images where the ROI and non-ROI parts are segmented from the original image. The ROI part represents the region more likely to contain landmines, whereas non-ROI is the less significant region.

#### 4.1.2. Results

This work utilized the U-Net segmentation model to improve the identification of landmines by analyzing satellite images. A dataset of 100 satellite photos from Google Earth was carefully selected, focusing on specific regions of interest (ROIs) intended for aerial surveys conducted using UAVs. The U-Net architecture was specifically designed for binary segmentation, allowing it to accurately determine the categorization of each pixel as either belonging to the ROI or the non-ROI category. To mitigate the limitations of the restricted dataset, the researchers implemented data augmentation techniques like scaling, rotation, shearing, flipping, and zooming, enhancing the model’s capacity to generalize well over a wide range of circumstances. During a comprehensive training and validation process, essential performance measures such as loss and accuracy provided vital insights into the model’s competency, depicted in Figure 9. The segmentation methodology has been developed to enhance the accuracy of UAV survey operations, enabling focused examination of areas that are more likely to harbor landmines.

### 4.2. Path Planning

Experiments for UAV-based surveys for different input locations have been performed using Python and ArduPilot-based simulation. It helps to check the efficiency of the obtained coverage path using the proposed method regarding path length, time, and energy consumption. In conjunction with the region’s geometry, certain UAV parameters govern the generation of coverage routes. Table 1 lists UAV parameters and their corresponding values. The drone speed determines the maximum speed of a drone to the ground. At the turns, the velocity can vary, based on the trajectory. Regarding coverage path generation, the UAV’s height is also crucial. The UAVs are outfitted with various sensors for data collection during aerial surveys. With an increase in height, the sensor footprint also changes, resulting in a modification of the required line spacing along the coverage path. The percentage of overlap can be selected for the coverage depending on the required resolution. The importance of performing surveys at lower altitudes is in their capacity to provide enhanced resolution and sensitivity, which are essential for the precise identification and localization of landmines. By operating at lower altitudes, UAVs can acquire more detailed data, mitigating the potential for erroneous negative results and augmenting the overall detection precision. This enhances the identification of tiny or deeply concealed landmines that may be disregarded in surveys conducted at higher altitudes. Several aspects, including altitude, sensor field of view, spatial resolution, and overlap percentage impact the survey line spacing in coverage route planning using UAVs. To ensure simplicity and facilitate implementation, the present research has used some fixed values listed in Table 1. Two types of coverage paths were considered. The first one with line space of 10 m, a UAV height of 5 m, and a maximum speed of 10 m/s. The second one is based on magnetometer-based sensing as employed in [14]. It has a line space of 3 m, with a UAV height of 1 m and a maximum UAV speed of 5 m/s.

The integration of digital elevation models (DEMs) plays a crucial role in the aerial operations of UAVs. The provided datasets enable a thorough depiction of terrain elevation, enabling an enhanced understanding of the topographical differences within the region. By incorporating DEMs into the UAV’s navigation system, the aircraft can make informed decisions in real time, adjusting its height and route to accommodate the current terrain conditions. In the context of our simulation configuration, we used the functionalities offered by the integrated DEM given by Mission Planner. It enables a dynamic representation of the terrain’s elevation data, revealing the topographical characteristics of the surveyed area. By incorporating the DEM data into our simulation, we can simulate real-world conditions and allow the UAV to alter its flight parameters in response to variations in terrain elevation.

The time a UAV requires to cover a given area is critical in many applications, such as search and rescue operations or disaster response efforts. The faster the UAV can cover the area, the quicker it can provide crucial information or aid to those in need. Minimizing energy usage can increase the flight time and distance a UAV can cover before it returns for recharging or refueling. It can improve the overall efficiency of the UAV mission and reduce costs. The energy consumption of a UAV increases with the distance covered. A longer coverage path will require the UAV to fly for a longer period, reducing the available flight time. Figure 10 depicts the total length of the path generated by various coverage path planning techniques. The spiral pattern generates a short path for simple regions, but with increased complexity, the path length becomes higher than the zig-zag. The proposed coverage pattern in the study was compared with the minTraversal strategy [22]. The minTraversal strategy produces a coverage trajectory that is longer than that generated by the zig-zag strategy.

The number of turning points in a UAV coverage path can significantly impact the energy consumption and flight time of the UAV. The turning point requires the UAV to slow down, change direction, and accelerate again, which consumes additional energy. The comparison of the number of turning points in different patterns is presented in Figure 11. The zig-zag pattern generally has fewer turning points but requires more sharp turns if the region is complex, and remains relatively smooth in the spiral pattern. Similarly, the time consumption is depicted in Figure 12. The zig-zag pattern generates the trajectory requiring lesser time to cover the entire region as compared to other methods.

Figure 13 shows the relative energy requirements of various coverage patterns. Similarly, the energy consumption per kilometer is depicted in Figure 14. Table 2 summarizes the coverage results obtained for different strategies. The zig-zag path planning approach has demonstrated superior performance, showcasing its potential to enhance efficiency and optimize UAV-based surveys. Through its innovative navigation strategy, the zig-zag pattern significantly reduces the number of turns required during the survey, resulting in a streamlined flight trajectory. This reduction in the number of turns translates to both time and energy savings, allowing the UAV to cover the designated area more swiftly and with reduced battery consumption. In addition, the zig-zag path planning method generates shorter survey routes than other methods. By strategically alternating flight directions, the zig-zag pattern ensures that adjacent survey lines are closely spaced, effectively reducing coverage gaps and enhancing the overall quality of data collection. The compact path design provides extensive coverage without compromising data precision.

## 5. Conclusions

The landmines continue to create problems for people residing in contaminated areas, and manual detection methods are time-consuming and risky. The automation of the detection process using UAVs can greatly help in the efforts of humanitarian demining. The work conducted is focused on the task of automating the path determination for UAV-based minefield surveys. It works by segmenting the required area and then generating the coverage path. It can help optimize the UAV performance while operating in complex terrains. The study compared the time and energy consumption of different coverage path planning algorithms. The results showed that a carefully planned zig-zag path was able to cover a larger area in a smaller amount of time with lesser energy consumption compared to other methods.

The study can be extended with real-time experiments where various factors for controlled and safe UAV operation can be explored under different environments. The use of multiple UAVs to further enhance the minefield coverage in various regions can be considered in future work. Moreover, dynamic UAV operation in the presence of obstacles and other environmental factors needs to be examined to ascertain robustness in UAV-based landmine detection.

## Figures and Tables

**Figure 1 sensors-23-07264-f001:**
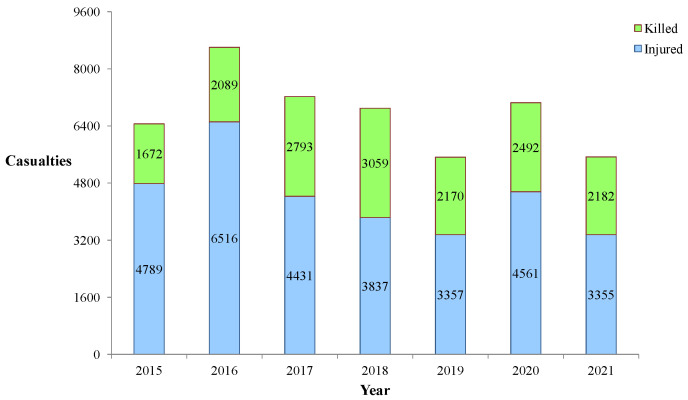
Casualties due to landmines/ERW during 2015–2021.

**Figure 2 sensors-23-07264-f002:**
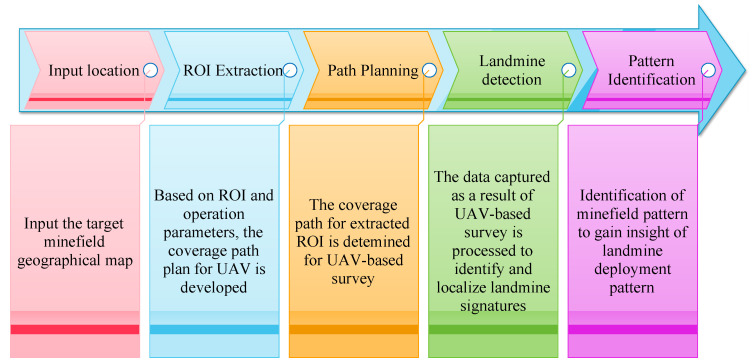
UAV-based landmine detection.

**Figure 3 sensors-23-07264-f003:**
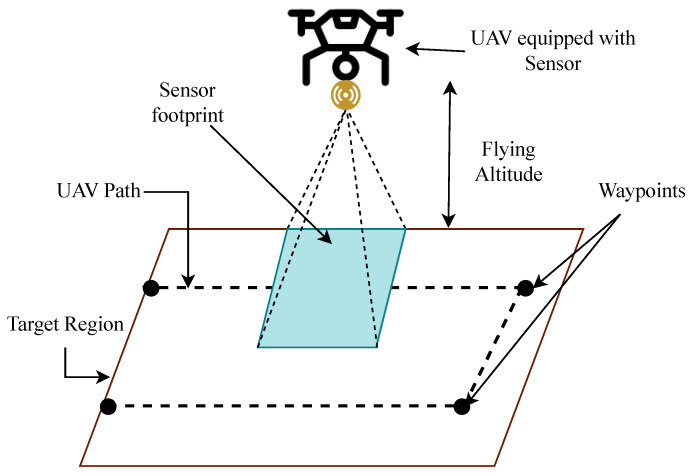
Representation of the UAV-based survey of the target area.

**Figure 4 sensors-23-07264-f004:**
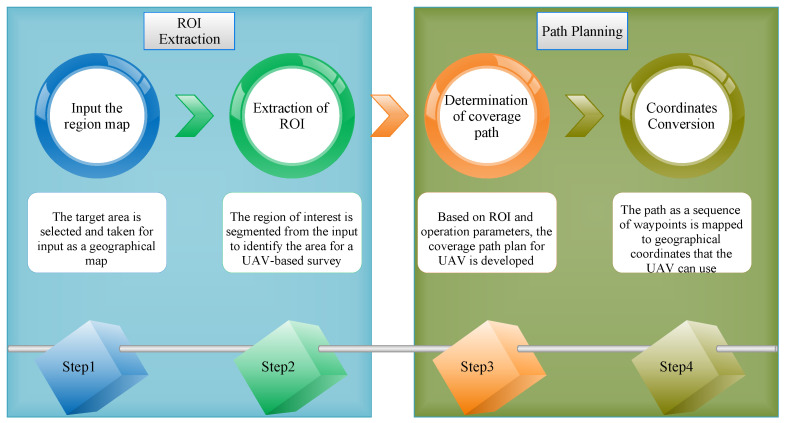
ROI segmentation and coverage path panning for UAV-based survey.

**Figure 5 sensors-23-07264-f005:**
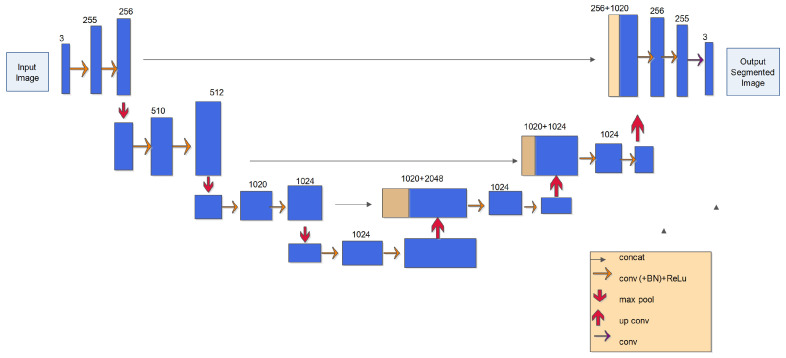
U-net architecture (example for 32 × 32 pixels in the lowest resolution).

**Figure 8 sensors-23-07264-f008:**
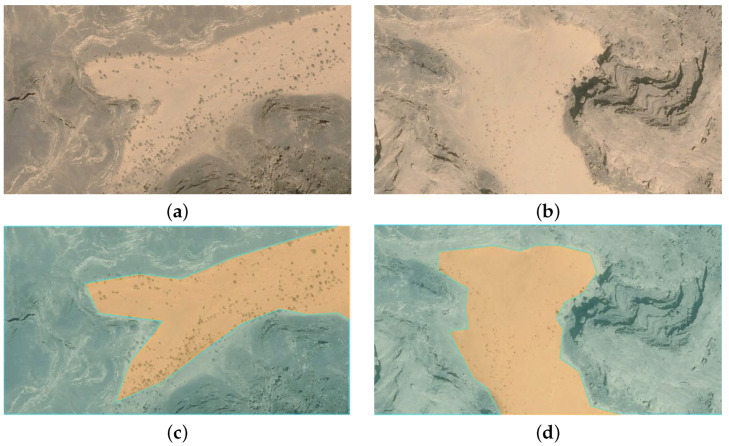
(**a**,**b**) Images taken from Google Earth; (**c**,**d**) are labeled images.

**Figure 9 sensors-23-07264-f009:**
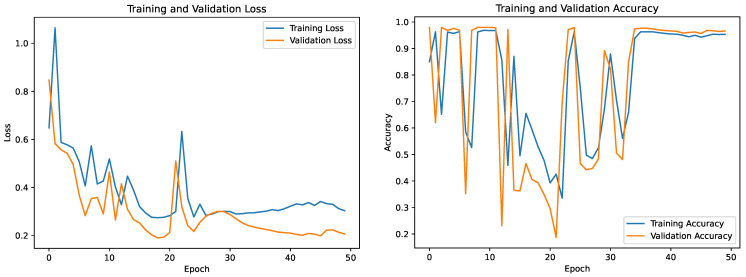
Graph showing the performance of the U-net model in terms of loss and accuracy on the given dataset.

**Figure 10 sensors-23-07264-f010:**
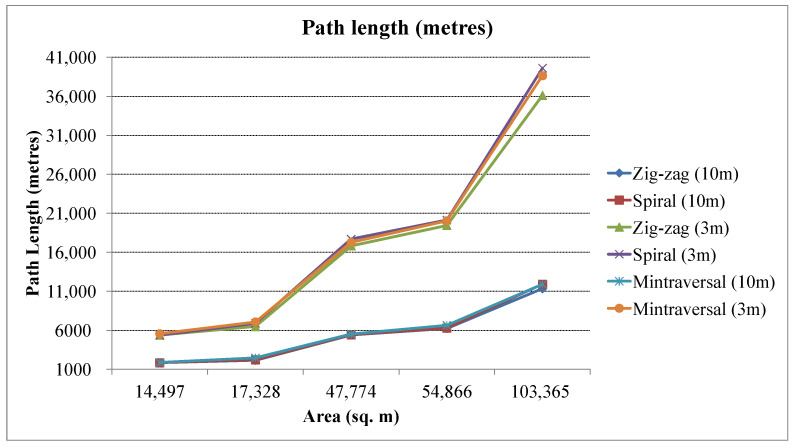
Total distance or path length (in meters) for different coverage paths.

**Figure 11 sensors-23-07264-f011:**
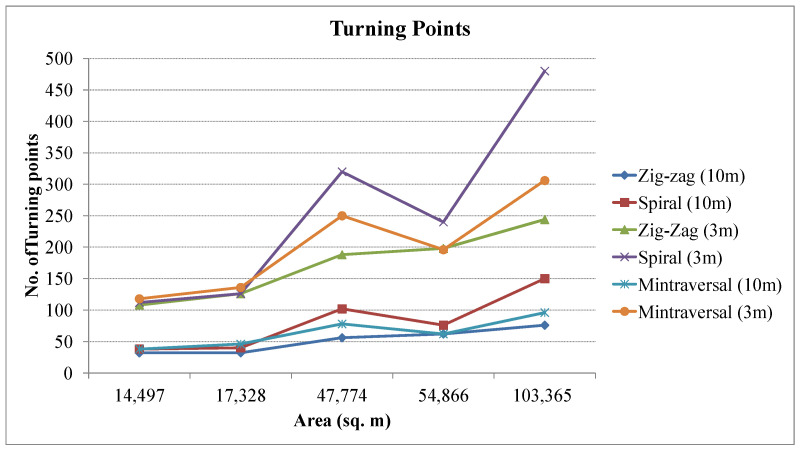
Number of turning points in different coverage paths.

**Figure 12 sensors-23-07264-f012:**
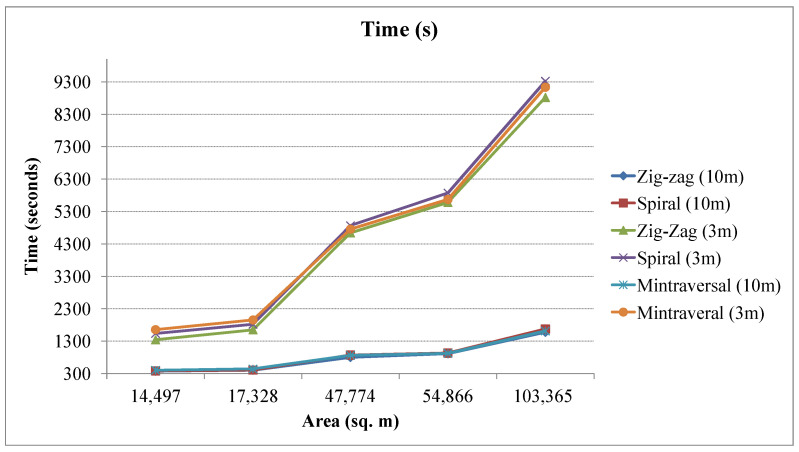
Total time consumption (in seconds) by UAV to traverse different coverage paths.

**Figure 13 sensors-23-07264-f013:**
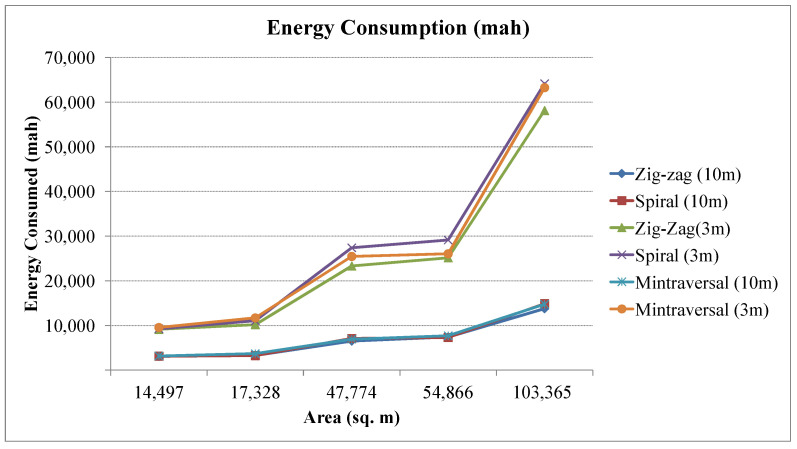
Energy consumption in milliampere-hour (mAh) for different coverage paths.

**Figure 14 sensors-23-07264-f014:**
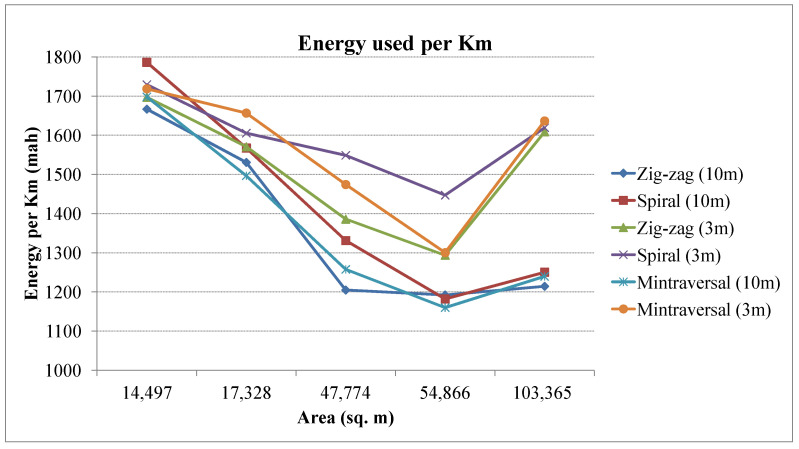
Energy consumed per kilometer for different coverage paths.

**Table 1 sensors-23-07264-t001:** Parameters considered for testing the coverage paths for UAVs.

Parameter	Value
UAV Max speed	5 m/s, 10 m/s
Altitude	1 m, 5 m
Line Gap	3 m, 10 m
Overlap	0.5, 0.5

**Table 2 sensors-23-07264-t002:** Comparative analysis of different coverage paths for various sample regions with line gap 10 m.

Sample	Area	Path	Path Length	Time(s)	Energy Consumed	Energy mAh/Km	No. of Turning Points	Average Speed
S1	14,497	Zig-zag	1849.64	385	3083	1666.81	32	4.8
S2	17,328	Zig-zag	2154	410	3298	1530.78	32	5.32
S3	47,774	Zig-zag	5405.41	801	6514	1205.09	56	6.75
S4	54,866	Zig-zag	6235.49	915	7435	1192.37	62	6.81
S5	103,365	Zig-zag	11,350.54	1575	13,785	1214.47	76	7.21
S1	14,497	Spiral	1750.99	385	3128	1786.42	38	4.55
S2	17,328	Spiral	2223.95	405	3264	1467.66	40	5.42
S3	47,774	Spiral	5334.95	872	7103	1331.41	102	6.12
S4	54,866	Spiral	6295.56	900	7317	1162.25	76	6.99
S5	103,365	Spiral	11915	1675	14,898	1250.36	150	7.11
S1	14,497	minTraversal	1876.74	405	3187	1698.15	38	4.63
S2	17,328	minTraversal	2473.92	445	3703	1496.81	46	5.56
S3	47,774	minTraversal	5522.08	865	6945	1257.67	78	6.38
S4	54,866	minTraversal	6645.74	925	7710	1160.14	62	7.18
S5	103,365	minTraversal	11,901.89	1615	14,756	1239.80	96	7.36

## Data Availability

The data are contained within the article.

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
