# Peer review of "Unmanned Ariel Vehicle (UAV) Path Planning for Area Segmentation in Intelligent Landmine Detection Systems"

_sensors, 2023, doi:10.3390/s23167264_

Round 1
Reviewer 1 Report
Based on Fig 9, the model training may need to consider the complicated data augmentation to increase the samples.
Author Response
Comment 1: Based on Fig 9, the model training may need to consider the complicated data augmentation to increase the samples.
Response: The authors thank the reviewer for insightful feedback. The dataset contains the satellite images which we segmented. The data augmentation on the images has been done to increase the samples, which, in turn, changes the experimental results of the training model. The data augmentation is done by rotating the images, horizontal flip, zooming , height, and width shift. The details and updated graphs have been added in the revised manuscript section 4.1.2, Figure 9.
Reviewer 2 Report
In this manuscript, the authors investigate the area segmentation and path planning problems of UAVs for landmine detection. Specifically, a deep learning-based extraction method is used to isolate the region of interest from the satellite imagery, and then a path planning problem is solved for UAV-based coverage.
Using UAVs for landmine detection has great importance and interest to the UAV community for both military and civilian applications. The topic investigated in this manuscript fall in the scope of this journal.
One major concern is about the landmine detection precision using sensors onboard a UAV. What factors affect the detection accuracy? Does it depend on the flight altitude and speed of the UAV? For example, what sizes of landmines can be detected under what soil conditions at what UAV altitude and speed? Should these factors be considered in UAV path planning?
Section 3.1: What’s new about the segmentation method used by the authors and why it is novel and superior to other existing ones? More discussion is needed.
Section 3.2: Spiral path planning and zig-zag path planning algorithms are used without clear justification. Why are these algorithms not others used in this work given the extensively explored UAV path planning area? The novelty of these algorithms needs further explanation.
It is not clear to me how the time and power consumption of the UAV are considered in the problems solved in this manuscript.
It is unclear to me which part of the proposed approach is implemented onboard the UAV and what will be done offline on the ground. Also, are the onboard implemented computations within the capability of the current UAV flight hardware and energy limit of onboard battery? More discussion and analysis of the computational complexity and time consumption of the algorithms are needed in the Experimentation and Results section.
Author Response
Comment 1: One major concern is about the landmine detection precision using sensors onboard a UAV. What factors affect the detection accuracy? Does it depend on the flight altitude and speed of the UAV? For example, what sizes of landmines can be detected under what soil conditions at what UAV altitude and speed? Should these factors be considered in UAV path planning?
Response: The authors thank the reviewer for insightful feedback. The accuracy of landmine detection using onboard sensors is influenced by a combination of factors. These factors include sensor technology, resolution, sensitivity, flight altitude, and speed of the UAV, among others. The specific characteristics of the landmine, such as its size, depth, and composition, also play a significant role in detection accuracy. The revised manuscript now places an emphasis on these factors. For the sake of simplicity, our proposed approach entails generating a generalized path that adheres to predetermined line gap and UAV height specifications. This streamlined approach facilitates ease of implementation and serves as a foundation for more complex adaptations in the future. The required details have been added in the revised manuscript (Section 1).
Comment 2: Section 3.1: What’s new about the segmentation method used by the authors and why it is novel and superior to other existing ones? More discussion is needed.
Response: The authors thank the reviewer for the valuable comment. The segmentation model here we used was U-Net. We used the semantic segmentation model for our dataset because we took satellite images of a particular region. U-Net model produced masks with separate labels for each land cover type i.e., ROI and non-ROI. Also, U-Net architecture was built to handle small amounts of data. Due to these reasons, we prefer the U-Net model. The details have been added to the revised manuscript in Section 3.1.
Comment 3: Spiral path planning and zig-zag path planning algorithms are used without clear justification. Why are these algorithms not others used in this work given the extensively explored UAV path planning area? The novelty of these algorithms needs further explanation.
Response: The authors thank the reviewer for the valuable comment. The choice of these algorithms was made after careful consideration of their suitability for UAV-based landmine detection missions. The spiral path planning algorithm was chosen due to its ability to efficiently cover large areas while maintaining a focus on the center, which aligns well with the need for comprehensive coverage in landmine detection. The zig-zag algorithm is chosen for its unique ability to minimize the number of turns and overall path length, which are crucial factors in optimizing the efficiency of UAV-based aerial surveys. While other path planning algorithms have been explored, our study aimed to provide a focused and in-depth analysis of these specific algorithms in the context of landmine detection. The novelty of our work lies in the tailored adaptation and evaluation of these algorithms for this critical application. We have revised the manuscript to provide a more detailed explanation of our algorithm selection process to enhance the clarity of our methodology. (Section 3.2.1)
Comment 4: It is not clear to me how the time and power consumption of the UAV are considered in the problems solved in this manuscript.
Response: The authors thank the reviewer for the valuable comment. In our study, we have focused on optimizing coverage path planning for landmine detection by emphasizing the minimization of path length and turning points. While these factors were not explicitly discussed in terms of time and power consumption in the initial manuscript, they directly contribute to operational efficiency. The details have been added to the revised manuscript. (Section 3.2.1)
Comment 5: It is unclear to me which part of the proposed approach is implemented onboard the UAV and what will be done offline on the ground. Also, are the onboard implemented computations within the capability of the current UAV flight hardware and energy limit of onboard battery? More discussion and analysis of the computational complexity and time consumption of the algorithms are needed in the Experimentation and Results section
Response: The authors sincerely appreciate your thorough review of our manuscript. In our study, the path is generated offline, based on specified parameters such as UAV altitude, sensor coverage, and desired line gap. The generated path is subsequently transferred to the UAV's flight control system for execution during the mission. The onboard computations primarily pertain to real-time navigation and control, ensuring that the UAV accurately follows the predetermined path. The information has been updated in the revised manuscript (Section 1). The computational complexity of the algorithms has been updated in Section 3.2.1.
Reviewer 3 Report
The paper proposes a method for autonomous UAV-based landmine detection. Analyzing the paper, I identified the following issues:
1. The link between landmine detection and the proposed method is very weak. The method is a general one and with limited novelty (this method is well-known in scientific literature). I suggest further adapting the proposed method to the characteristics of landmine detection sensors.
2. The segmentation process is confusingly presented. How exactly the segmentation is done? The use of Google Earth photographs in this process is also confusing. How exactly the landmine-affected areas are identified in these photos? Some details are needed.
3. There are some typos: line 222-there is no reference to Choi et al.; lines 225-226 – it is written “in et al..”; line 280 – it is written “Path” instead of “path”, etc. Please check the entire manuscript.
4. How a 2D simulation (fixed altitude of 5m) fits with real-life circumstances where the terrain may be complex and may change abruptly?
5. On what basis the “line gap” (10m) was chosen? Please note that you have already chosen an altitude of 5 m. Are these values in line with landmine detection sensors? I suggest recalibrating the simulation to a specified landmine detection sensor.
6. A comparison with other landmine detection methods may be beneficial.
English needs some polishing. There are some typos to correct. Please check the entire manuscript.
Author Response
Comment 1: The link between landmine detection and the proposed method is very weak. The method is a general one and with limited novelty (this method is well-known in scientific literature). I suggest further adapting the proposed method to the characteristics of landmine detection sensors.
Response: The authors are thankful to the reviewer for the helpful feedback on our manuscript. While we recognize that the core principles of path planning are established in the scientific literature, the novelty of our work resides in the specific adaptation and application of these principles to the domain of UAV-based landmine detection. In response to your valuable suggestion, we have taken the initiative to refine our approach in the revised manuscript. Specifically, we have extended our coverage path generation methodology to align with the specific characteristics and requirements of landmine detection sensors. The consideration has been given to coverage path with line spacing 3m at height 1m as used in existing literature for magnetometer-based survey. The same has been updated in output graphs. (Section 4.2, Fig 10-14)
Comment 2: The segmentation process is confusingly presented. How exactly the segmentation is done? The use of Google Earth photographs in this process is also confusing. How exactly the landmine-affected areas are identified in these photos? Some details are needed.
Response: The authors are grateful for the reviewer’s evaluation. We needed the dataset containing images of a region where we assumed landmines were buried for UAV Path Planning for an Intelligent Landmine Detection System. For that we took Google Earth images of specific areas with GPS coordinates. The aim is to segment these images to into two classes: ROI and non-ROI. The ROI part is the region more likely to contain landmines, while non-ROI region may be rocks, mountains, and buildings. So ROI is the one desirable for UAV-based survey. We have used the U-Net model for the segmentation of these satellite images. The model is trained on the dataset. The trained model can extract the desired region from the input satellite image which is later used for planning the coverage path for UAV. The required details have been updated in the revised manuscript. (Section 4.1.1)
Comment 3: There are some typos: line 222-there is no reference to Choi et al.; lines 225-226 – it is written “in et al..”; line 280 – it is written “Path” instead of “path”, etc. Please check the entire manuscript.
Response: The authors express gratitude to the reviewer for their thoughtful evaluation. In light of the comment, we have conducted a thorough review of the entire manuscript to rectify the mentioned typos and ensure the accuracy and clarity of our references and text.
Comment 4: How a 2D simulation (fixed altitude of 5m) fits with real-life circumstances where the terrain may be complex and may change abruptly?
Response: The authors thank the reviewers for the insightful comment. In our simulation setup, we have harnessed the capabilities of the inbuilt digital elevation model (DEM) provided by Mission Planner. This tool facilitates a dynamic representation of the terrain's elevation data, offering insights into the topographical characteristics of the surveyed area. By incorporating this DEM data into our simulation, we emulate real-world conditions and enable the UAV to adapt its flight parameters in response to changes in terrain elevation. The required details have been incorporated in the updated manuscript. (Section 4.2)
Comment 5: On what basis the “line gap” (10m) was chosen? Please note that you have already chosen an altitude of 5 m. Are these values in line with landmine detection sensors? I suggest recalibrating the simulation to a specified landmine detection sensor.
Response: The authors extend our gratitude for your thoughtful consideration and inquiry regarding the parameters chosen in our study. In the revised manuscript, we have included a discussion addressing the necessity of adapting parameters, including the line gap and UAV altitude, to match the specifications of landmine detection sensors. We have also adopted a survey height of 1 meter and a line gap of 3 meters, in line with magnetometer-based surveys. The same has been updated in output graphs. (Section 4.2, Fig 10-14)
Comment 5: A comparison with other landmine detection methods may be beneficial.
Response: The authors genuinely appreciate the insightful suggestion regarding the inclusion of a comparison with other landmine detection methods in our study. In the revised manuscript the Mintraversal algorithm has indeed been utilized for comparison in our study. The obtained results have been added and the output graphs have been updated in the revised manuscript. (Section 4, Table2, Fig 10-14 )
Comment 6: English needs some polishing. There are some typos to correct. Please check the entire manuscript.
Response: The authors sincerely appreciate your diligent review of our manuscript. The feedback on the English language quality and the identification of typos are invaluable contributions to improving the overall readability and professionalism of our work. In response to your comment, we have undertaken a comprehensive review of the entire manuscript to rectify the mentioned typos and grammatical errors.
Round 2
Reviewer 2 Report
Thank you for addressing my comments.
Reviewer 3 Report
The authors have successfully solved all y comments/concerns.
English is fine.